# Spatiotemporal Variations of Meteorological Droughts and the Assessments of Agricultural Drought Risk in a Typical Agricultural Province of China

**Mengjing Guo [1], Jing Li [1,\*], Yongsheng Wang [2], Qiubo Long [3] and Peng Bai [4]**

1   State Key Laboratory of Eco-hydraulics in Northwest Arid Region, Xi'an University of Technology, Xi'an 710048, China; guomengjing@xaut.edu.cn
2   Key Laboratory of Regional Sustainable Development Modeling, Chinese Academy of Sciences, Beijing 100101, China; wangys@igsnrr.ac.cn
3   Hunan Hydro and Power and Design Institute, Changsha 410007, China; longqiubo17@126.com
4   Key Laboratory of Water Cycle and Related Land Surface Process, Institute of Geographic Sciences and Natural Resources Research, Chinese Academy of Sciences, Beijing100101, China; baip@igsnrr.ac.cn
\*   Correspondence: lijing8615@163.com

**Abstract:** Drought is one of the most common natural disasters on a global scale and has a wide range of socioeconomic impacts. In this study, we analyzed the spatiotemporal variations of meteorological drought in a typical agricultural province of China (i.e., Shaanxi Province) based on the Standard Precipitation Evapotranspiration Index (SPEI). We also investigated the response of winter wheat and summer maize yields to drought by a correlation analysis between the detrended SPEI and the time series of yield anomaly during the crop growing season. Moreover, agricultural drought risks were assessed across the province using a conceptual risk assessment model that emphasizes the combined role of drought hazard and vulnerability. The results indicated that droughts have become more severe and frequent in the study area after 1995. The four typical timescales of SPEI showed a consistent decreasing trend during the period 1960–2016; the central plains of the province showed the most significant decreasing trend, where is the main producing area of the province's grain. Furthermore, the frequency and intensity of drought increased significantly after 1995; the most severe drought episodes occurred in 2015–2016. Our results also showed that the sensitivity of crop yield to drought varies with the timescales of droughts. Droughts at six-month timescales that occurred in March can explain the yield losses for winter wheat to the greatest extent, while the yield losses of summer maize are more sensitive to droughts at three-month timescales that occurred in August. The assessment agricultural drought risk showed that some areas in the north of the province are exposed to a higher risk of drought and other regions are dominated by low risk.

**Keywords:** SPEI; meteorological drought; drought evolution; drought risk assessment

## 1. Introduction

Drought is one of the most widespread natural hazards worldwide. It has tremendous economic and social impacts, affecting everything from agriculture and water supply to public health and economic development. According to a report by Food Agriculture Organization (FAO) [1], drought ranks first among weather-related disasters in terms of economic impacts. Agricultural economic losses caused by droughts in developing countries are as high as 29 billion dollars per year between 2005 and 2015. Beyond direct economic impacts, droughts can threaten drinking water supplies and

ecosystems, and can even contribute to increased food prices and affect food security. Unlike natural hazards with more immediate impacts (e.g., floods, landslides, and wildfires), drought is a creeping phenomenon that develops slowly and can persist for years. Related impacts develop slowly and can linger for long times after the end of the drought. In the context of climate change, droughts are expected to become more frequent and severe in many regions around the globe during this century [2]. Despite climate projections are accompanied by large uncertainties, current research generally agreed that global warming may exacerbate the onset of drought, making the drought more intense and last longer in the 21st century [3]. The cases of perceived severe drought in the 2000s are abundant, such as the record-breaking drought in California (2012–2016), the millennium drought in Australia (2001–2009), and the extreme drought in Southwest China (2009–2010). These extreme drought events have received much attention from the media and scientific communities [4–6].

Droughts are among the most complex natural hazards and involve complex interactions between multiple influencing factors. Although the precipitation deficits are the primary cause of drought, other factors (e.g., soil texture, high temperature, poor water management, and soil erosion) can also cause or enhance droughts. Thus, characterizing drought requires a variety of geophysical factors, such as precipitation, temperature, soil moisture, and river flow; the choice of factors depends on the impact on society and environment [7]. Consequently, it is difficult to have a universal definition of drought. In general, drought definitions can be broadly categorized into four types: meteorological drought, hydrological drought, agriculture drought, and socioeconomic drought.

Each type of drought includes multiple indicators to quantify a drought. The choice of indicators usually depends on the purposes of the user and hydroclimatic variables included, yet different indicators may result in a different ranking of drought severity [8]. The commonly used meteorological drought indicators include the Palmer drought severity index (PDSI) [9], the standardized precipitation index (SPI) [10], and the standardized precipitation evapotranspiration index (SPEI) [11]. Each index has its own advantages and limitations. Some studies have well summarized these widely used drought indicators [12–14]. The PDSI was developed originally as an agricultural monitoring tool and has long been as a landmark in the development of drought indicators [15]. It quantifies monthly accumulated moisture anomalies based on the primitive soil water balance that incorporates precipitation, runoff, and evaporative demand into a water balance framework [11]. However, a major drawback of the PDSI is that PDSI values are not comparable between diverse climate regions because of the empirical constants used in the calculation of the PDSI [16]. The SPI is a widely used index to characterize meteorological drought due to its simplicity and low data requirements. The main limitation of the SPI is that it describes drought based only on precipitation data and does not consider other factors that influence droughts, such as temperature, wind speed, and soil water holding capacity [11]. In many cases, drought is caused by an imbalance between surface water input and output. To avoid the SPI limitations, the SPEI was developed by Vicente-Serrano et al. (2010), which is a multi-scalar index that provides a relatively flexible approach to describe the integrated effects of precipitation (P) and evaporative demand (i.e., potential evapotranspiration, and PET) on drought. In addition, the SPEI only includes a simple climatic water balance (D = P − PET), so it cannot provide the soil moisture status directly like PDSI [17,18]. Nevertheless, many studies have reported that the SPEI performs better than other indicators in identifying drought impacts on agriculture at regional and global scales [19–21]. For example, Labudová, et al. [19] reported that the SPEI reaches the higher correlation with the yield of maize than the SPI in Slovakia. Vicente-Serrano, *et al.* [22] found that the SPEI performs better than SPI and PDSI in identifying drought impacts on hydrological, agricultural, and ecological response variables at the global scale.

Drought risk assessment is essential for agricultural and water resources departments, which provides a framework for assessing the socioeconomic impacts of drought and helps to develop adaptive measures to reduce impacts of drought. Drought risk assessments require consideration of multiple factors, including drought hazard, as well as drought vulnerability and risk because droughts are not only a natural phenomenon but also a socioeconomic phenomenon. Moreover, drought risk assessments

need to be sector-specific since hazard and vulnerability of drought vary between economic sectors (e.g., agriculture, public water supply, and public health). Here, we only focus on the assessment of agricultural drought risk. Some studies have evaluated agricultural drought risk in China [23–25]. These studies provide valuable insights into where agricultural production is more likely to be affected. However, most assessments were performed at the country level [24] or the catchment scale [25], which may not reflect the details of drought risk because of high heterogeneity in environmental factors used in risk assessment. Moreover, some assessments were performed based on the SPI [24,25], which may underestimate the risk of agricultural drought under global warming since this index cannot reflect the impact of warming on drought evolution.

The main objectives of this paper include: (1) investigation of the drought evolution in a typical agricultural province of China (i.e., Shaanxi Province) using the SPEI during the period 1960–2016; (2) analysis of the responses of the crop yield (i.e., winter wheat and summer maize) to different time-scales of drought through a correlation analysis between the SPEI and the standard yield residual series (SYRS); and (3) evaluation of the agricultural drought risk based on a conceptual risk assessment approach (see Section 2.3). The findings of this study can help us to understand the spatiotemporal variations of droughts in Shaanxi Province better and provide guidance for taking measures to reduce the drought impacts on agriculture.

## 2. Materials and Methodology

### 2.1. Study Area and Data

Shaanxi Province consists of ten prefecture-level cities and covers an area of about 205,600 km$^2$ with about 37 million people (Figure 1). It is one of the important cradles of the Chinese civilization and 14 dynasties have estimated their capital here, such as Qin and Tang dynasties. Shanxi Province has long been as one of the main producing areas of China's grain. The annual double-crop rotation system, winter wheat and summer maize, is the most popular planting pattern in the province. The planting area of winter wheat and summer maize accounts for more than 70% of the province's cultivated land area, and their output accounts for about 85% of the province's total grain output [26]. The terrain of Shaanxi Province is high in the north and south and low in the middle. The northern part of the province is being part of the Ordos Desert, the Loess Plateau in the central part of the province, accounting for 40% of the province's land area, and the Qin Mountains running east to west in the south-central part. Due to its large space in latitude, the province spans multiple climates, including cold, cold semi-arid, and the humid subtropical climate, with the aridity index (PET/P) ranging from 0.62 to 3.45 (Figure 1b). The main land-use types of the province are grasslands (39.6%), croplands (31.9%), and woodlands (24.4%), and the croplands are mainly located in the plain area of central Shaanxi Province (Figure 1c).

Daily meteorological data from 42 stations during the period 1960–2016 were collected from the Chinese Meteorological Administration and the collected data include precipitation, temperature, relative humidity, wind speed, and sunshine hours. We also performed a rigorous quality inspection and control for meteorological data to eliminate the impact of non-climatic factors. The anusplin, a professional meteorological interpolation software that accounts for the influence of the topography on interpolated variables [27], was used to interpolate the gauge-based observations into grid-based data with a spatial resolution of 0.05° × 0.05°. The PET was calculated using the FAO-56 Penman-Monteith method, which has been recommended by FAO experts as the sole method for determining PET [28]. The annual yield data of winter wheat and summer maize in Shaanxi Province from 1989 to 2016 were obtained from the Shaanxi Province Bureau of Statistics [29]. It should be noted that the growing season of the winter wheat in Shaanxi Province is usually from October of the previous year to June of the following year; while the growing season of the summer maize is usually from June to October of the current year. In addition, global soil water-holding capacity data and global irrigation area data were also employed to assess the agriculture drought risk. The resolution of the two datasets is 1/12° × 1/12° and they are available at [30] and [31].

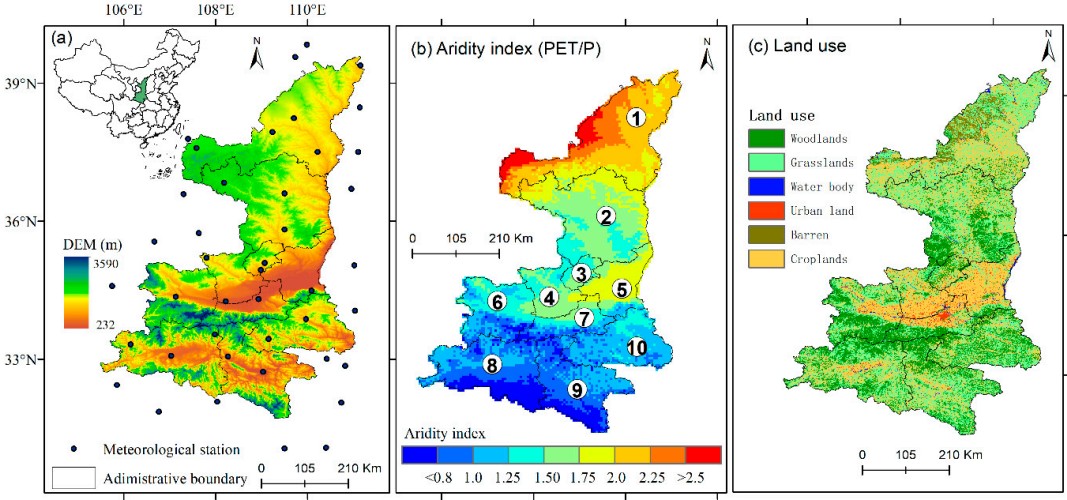

**Figure 1.** The location and terrain of the study area (**a**) and the spatial patterns of aridity index (**b**) and land use (**c**) across the study area. The study area includes ten prefecture-level cities (see Figure 1b), namely Yulin (①), Yan'an (②), Tongchuan (③), Xianyang (④), Weinan (⑤), Baoji (⑥), Xi'an (⑦), Hanzhong (⑧), Ankang (⑨), and Shanluo (⑩), respectively.

## 2.2. Calculation of SPEI and SYRS

Here, the SPEI was used for characterizing drought evolution in the study area during the period 1960–2016. A key advantage of SPEI over other widely used drought indices is that the index considers the effects of evaporative demand on drought severity [11], which allows it to better reflect the evolution of drought under climate change. The SPEI runs at a monthly time step and requires the monthly P and PET as inputs. Similar to SPI, the SPEI must be associated with a specific timescale to reflect the impact of previous water deficits on current drought status [19,21,23,24]. The term drought timescale refers to the time lag that typically exists between the start of water shortage and the identification of its consequences [32]. For example, a value of SPEI at timescales of six months would imply that data from the current month and of the past five months will be used for computing the SPEI value for a given month. In general, the SPEI values at shorter timescales (e.g., one-month and three-month) reflect the short-term moisture conditions and often has a good correlation with soil moisture; while the SPEI values at longer timescales (e.g., six-month and 12-month) reflect the medium- and long-term water balance patterns and are usually correlated to river flows, reservoir storages, and groundwater levels [23]. In practice, the SPEI-03 is commonly used to characterize agricultural droughts since it is more correlated with soil moisture than other timescales of SPEI [19,32,33]. Table 1 provides a categorization of drought severity according to the SPEI, where the more negative the SPEI value corresponds to the more severe the drought. Detailed calculation procedure of SPEI can refer to Vicente-Serrano et al. (2010) [11] and a SPEI R package has been developed by authors to help to calculate SPEI time series and is available at [34].

**Table 1.** Drought categories based on the Standard Precipitation Evapotranspiration Index (SPEI) and categories of yield losses based on the SYRS.

| SPEI | Drought Category | SYRS | Category of Yield Losses |
|:---:|:---:|:---:|:---:|
| −1.0~1.0 | Normal | −0.5~0.5 | Normal |
| −1.5~−1.0 | Moderate drought | −1.0~−0.5 | Low yield losses |
| −2.0~−1.5 | Severe drought | −1.5~−1.0 | Moderate yield losses |
| ≤−2.0 | Extreme drought | ≤−1.5 | High yield losses |

Drought can cause significant yield reductions, particularly in rainfed agricultural systems. Here, the effect of meteorological droughts on crop yields was evaluated by a correlation analysis, i.e.,

calculating the Spearman's correlation coefficient between the detrended SPEI and crop yield anomaly time series during the growing season of crops. The collected crop yield data cannot be directly used to establish a drought–yield relationship because the change in crop yield is affected by several factors in addition to droughts, such as new management practices, advances in agricultural technology, grain price, and increased fertilizer application. To remove the effect of these non-climatic factors, the SYRS was utilized to perform the correlation analysis with the detrended SPEI time series. The SYRS eliminates the trend in annual yield time series but maintains the inter-annual variation of the yield time series [19,21,35]. Figure 2 shows the detrended yield time series of winter wheat and summer maize in the Shaanxi Province from 1989 to2016. Detailed calculation procedure of the SYRS can refer to Potopová et al. (2016) [21]. Categories of yield losses based on the SYRS are presented in Table 1.

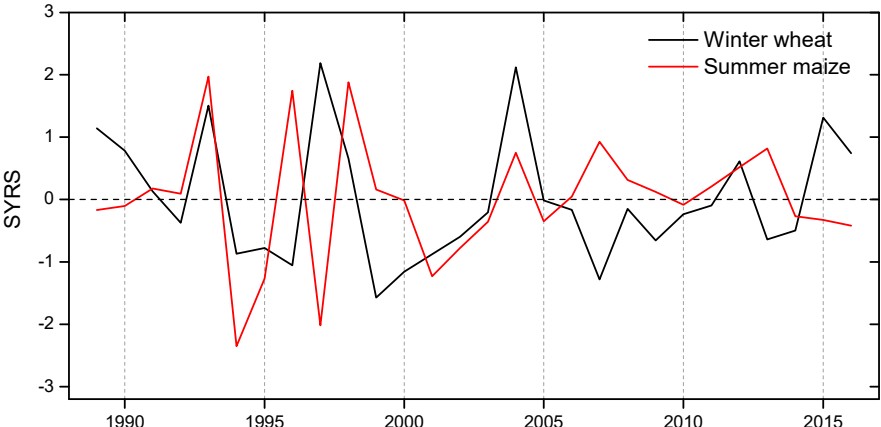

**Figure 2.** The inter-annual variability of standard yield residual series (SYRS) for winter wheat (black line) and summer maize (red line) in the Shaanxi Province.

### 2.3. Assessment of Agriculture Drought Risk

One of the main purposes of disaster risk assessment is to answer where it is more likely to be affected by disasters. Several approaches are available for assessing the risk of natural disasters. Among them, the conceptual models developed by UNDP (United Nations Development Program) were widely used [36], which defines risk by emphasizing the integrated function of physical hazards and the vulnerability of disasters to exposures [23,37]. Correspondingly, such risk assessment models consist of two parts: the assessment of hazard and the assessment of vulnerability. Here, we only focused on the assessment of agricultural drought risk. The conceptual models assess the agricultural drought risk by incorporating the hazard and vulnerability assessments [38,39]:

$$R(h, v) = f(h) \times f(v) \tag{1}$$

where $R$ is the agricultural drought risk; the function $f(h)$ denotes the agricultural drought hazard (see Equation (2)), and the function $f(v)$ denotes the agricultural drought vulnerability (see Equation (3)).

In general, the hazard was estimated as the probability of occurrence of a potentially damaging phenomenon [38]. Here, the frequency of SPEI at three kinds of drought severity (that is, moderate, severe, and extreme droughts) were used for quantifying the drought hazard [23–25]. The drought frequency for a specific drought severity was classified into four categories using the natural breaks approach. Each class of occurrences was assigned a rating from 1 to 4, and each drought severity category was assigned a specific weight from 1 to 3 (see Table 2). The higher rating corresponds to the higher frequency of occurrence, and the higher weight indicates the higher drought severity. Then, these weights and ratings were combined into a comprehensive drought hazard index (DHI), which is calculated as:

$$DHI = (MD_w \times MD_r) + (SD_w \times SD_r) + (VD_w \times VD_r) \tag{2}$$

where MD*w*, SD*w*, and VD*w* are the weight for moderate drought, severe drought, and extreme drought, respectively; MD*r*, SD*r*, and VD*r* are the ratings for moderate drought, severe drought, and extreme drought, respectively.

Vulnerability assessments are a key component of drought risk assessment as they support the design of mitigation actions to target sectors or more sensitive populations [39]. There are multiple definitions of vulnerability. Here, we adopted the following definition: the vulnerability is defined as the ability of those elements to respond to and recover from the impacts of such hazards [40]. Drought vulnerability is affected by numerous factors associated with socioeconomic and physical factors. Here, we selected the climatic factor, soil factor, and irrigation factor to construct the assessment model of drought vulnerability after systematic analysis of previous studies on drought vulnerability assessment [25,41–43]. The climatic factor directly determines the degree of water deficit under natural conditions during the crop growing season. Soil properties affect the response of crops to precipitation deficit: the plants living in soils with strong water-holding capacity is generally more drought tolerant under the same water deficit condition. The irrigation factor greatly alleviates the drought caused by precipitation deficit. The aridity index (AI, defined as the ratio of PET to P), available soil water-holding capacity (AWC), and the percentage of total area equipped for irrigation (IRR) were utilized to quantify the above three factors. The datasets of AWC and IRR can be obtained from the publicly available datasets (see Section 2.2) and the AI is calculated using the collected meteorological data. Each of the factors was divided into four classes and assigned weights from one to four (see Table 2). The three factors were combined into a composite index of agricultural drought vulnerability (DVI):

$$DVI = W_{AI} + W_{AWC} + W_{IRR} \tag{3}$$

where $W_{SCWD}$, $W_{AWC}$, and $W_{IRR}$ are the weights for AI, AWC, and IRR, respectively.

**Table 2.** Weight and rating scores based on the cumulative probability distribution of SPEI [24,41] and the used factors and their weights used for drought vulnerability assessment [42]. AI: aridity index, AWC: available soil water-holding capacity, IRR: the percentage of total area equipped for irrigation.

| Drought Intensity | Weight | Frequency (%) | Rating | Vulnerability Factor | Categories | Weight |
|---|---|---|---|---|---|---|
| Moderate | 1 | <9.0 | 1 | Climate (AI) | <1.0 | 1 |
|  |  | 9.0–10.0 | 2 |  | 1.0–1.5 | 2 |
|  |  | 10.0–11.0 | 3 |  | 1.5–2.0 | 3 |
|  |  | >11.0 | 4 |  | >2.0 | 4 |
| Severe | 2 | <3.5 | 1 | Soil (AWC, mm) | >250 | 1 |
|  |  | 3.5–4.5 | 2 |  | 175–250 | 2 |
|  |  | 4.5–5.5 | 3 |  | 100–175 | 3 |
|  |  | >5.5 | 4 |  | <100 | 4 |
| Extreme | 3 | <1.5 | 1 | Irrigation (IRR, %) | >50 | 1 |
|  |  | 1.5–2.0 | 2 |  | 25–50 | 2 |
|  |  | 2.0–2.5 | 3 |  | 5–25 | 3 |
|  |  | >2.5 | 4 |  | <5 | 4 |

Finally, the index of agricultural drought risk (DRI) was calculated as a multiplicative formula linking the DHI and DVI:

$$DRI = DHI \times DVI \tag{4}$$

The values of DRI is categorized into four groups based on the natural breaks method and the higher value indicates the larger agricultural drought risk.

## 3. Results and Discussions

### 3.1. Drought Patterns at Different Timescales

We analyzed drought patterns at the whole province scale over the period 1960–2016. The Hovmoller-type diagram (see Figure 3a) presents the temporal variation of drought and wet

episodes at a 1- to 12-month timescale. It can be seen that persistent droughts occurred in 1960–1965, 1995–2000, and 2015–2016, and the most severe drought episodes occurred in 2015–2016, lasting for several months. Drought/wet conditions contrasted sharply before and after 1995. Before 2005, the drought was characterized by short duration, low frequency, and intensity; the frequency and intensity of drought increased significantly after 1995. For example, on a six-month time scale (SPEI-06), there is no extreme drought before 1995, whereas it occurred four times after 1995 and the maximum duration reaches to six months (from July to December in 2015) (Figure 3d). Increased drought frequency and intensity since 1995 is likely due to the effects of climate change. With global warming, the Earth's climate system becomes more unstable than ever before [44], causing drought events to become more frequent in many regions of the world [45–47].

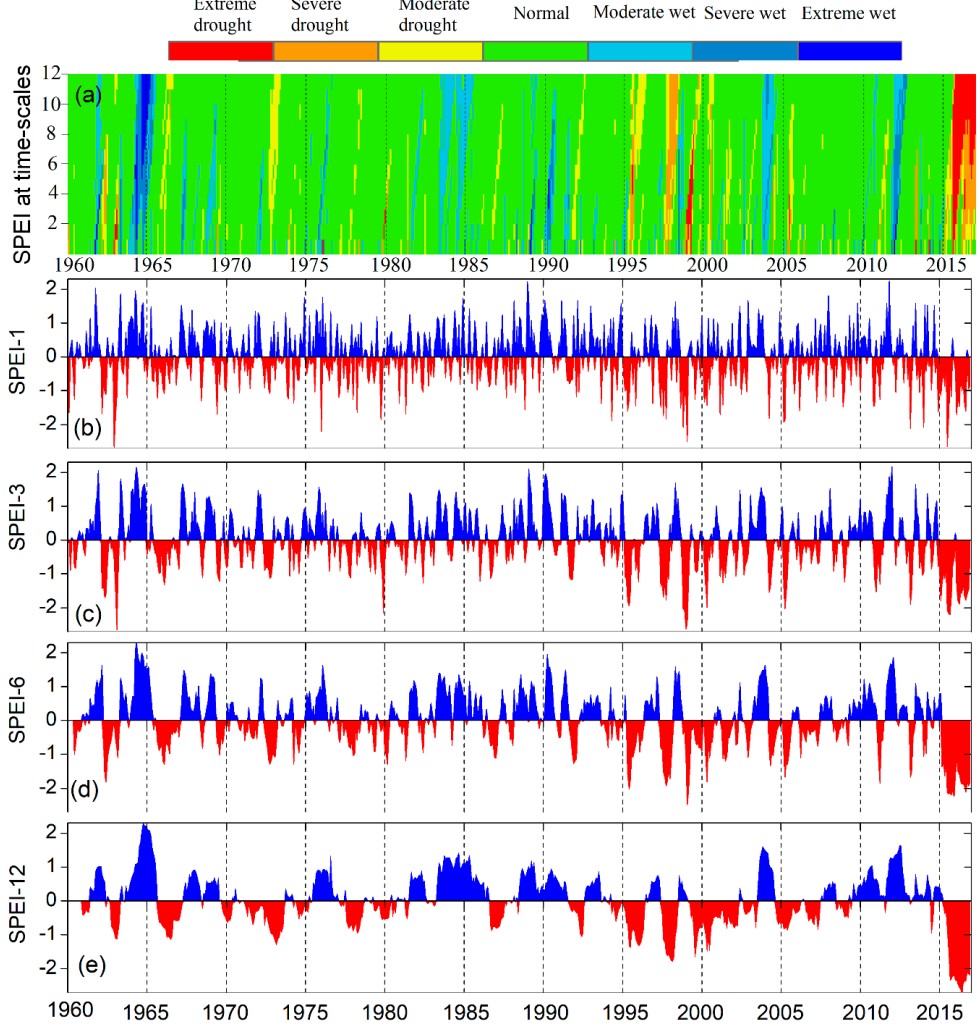

**Figure 3.** (**a**) Spatiotemporal evolution of SPEI series with 1- to 12-month lags from 1960 to 2016 and (**b–e**) temporal evolution of SPEI at 1-, 3-, 6-, and 12-month lags across the Shanxi Province.

The histograms (Figure 3b–e) show the temporal variations of SPEI at four typical timescales, i.e., 1-, 3-, 6-, and 12-month lags). The longer the timescale of SPET, the smoother the time series becomes since the values of climatic water deficit (D = P – PET) used in SPEI calculation are summed on the scale of interest. Thus, the long-term SPEI time series (e.g., SPEI-12) generally exhibits a longer duration than the short-term SPEI time series. As a result, drought characteristics identified by different timescales of SPEI are usually different from each other. For example, the most severe drought (corresponding to the lowest value in the SPEI time series) identified by the SPEI-01, SPEI-03, SPEI-06, and SPEI-12 occurred in December 1962 (–2.67), February 1963 (–2.65), February 1999 (–2.49), and June 2016 (–2.81), respectively.

### 3.2. Spatial Patterns in Drought Frequency and Drought Trends

We calculated the frequency in different intensities of drought at four typical timescales (i.e., 1-, 3-, 6-, and 12-month) across the study area (see Figure 4). Here, the frequency is defined as the number of months that a specific type of drought occurred as a percentage of the total number of the time series for the period of 1960–2016. The results indicate that the spatial pattern of drought frequency varies with timescales and types of drought. Even for the same type of drought, drought frequency exhibits a significant spatial difference at different timescales. For example, the severe drought characterized by SPEI-01 shows a higher occurrence frequency in the central part of Shaanxi Province, while the severe drought characterized by SPEI-12 has a higher frequency in the northern part of the province (Figure 4e,h). Here, we only analyzed the spatial pattern of drought frequency for SPEI-03 given that the SPEI-03 is generally considered to be better to reflect soil moisture dynamics than other timescales of SPEI. The moderate drought characterized by SPEI-03 shows a higher frequency in the south parts of the province (Figure 4b), while the severe drought has a higher occurrence of frequency in some areas in the northern (the junction areas of Yulin and Yan'an cities) and western parts (e.g., Baoji city) of the province. The extreme drought characterized by SPEI-03 is more likely to occur in the northern parts of the province (e.g., Yulin, Yan'an, and Tongchuan cities).

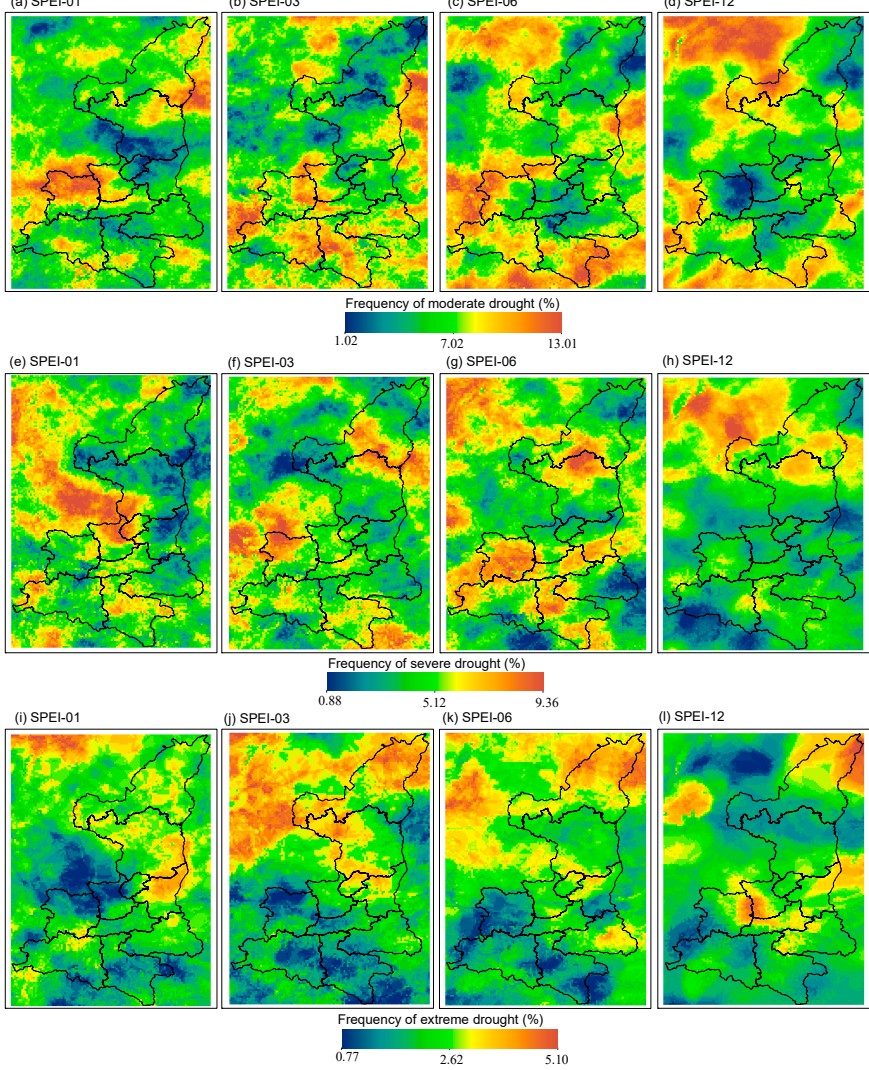

**Figure 4.** Spatial patterns of frequency in moderate drought months (a–d, $-1.0 \leq \text{SPEI} < -1.5$), severe drought months (e–h, $-2.0 < \text{SPEI} \leq -1.5$), and extreme drought months (i-l, $\text{SPEI} \leq -2.0$) across the study area.

Figure 5 illustrates the spatial patterns in the annual trend of SPEI at four typical timescales over the period of 1960–2016. The four typical timescales of SPEI demonstrates similar spatial patterns, although spatial differences among them do exist (Figure 5a–d). Droughts at four typical timescales are dominated by a decreasing trend. The most significant decreasing trend appears in the central part of the province (e.g., Baoji, Weinan, and Xi'an cities), where is the main producing area of the province's grain. At the whole province scale, the SPEI time series at four typical timescales presents a similar inter-annual variability over the period 1960-2016 (Figure 5e). The annual trend in the SPEI time series at 1-, 3-, 6- and 12-month lags is –0.05/10 year, –0.07/10 year, –0.09/10 year, –0.08/10 year, respectively. The most severe drought occurred in 2016, showing that the values of SPEI at four typical timescales are significantly less than the values in other years. An increased drought trend in Shaanxi Province has also been reported by Jiang et al. (2014) [48] and Zhou et al. (2014) [49] using different drought indices.

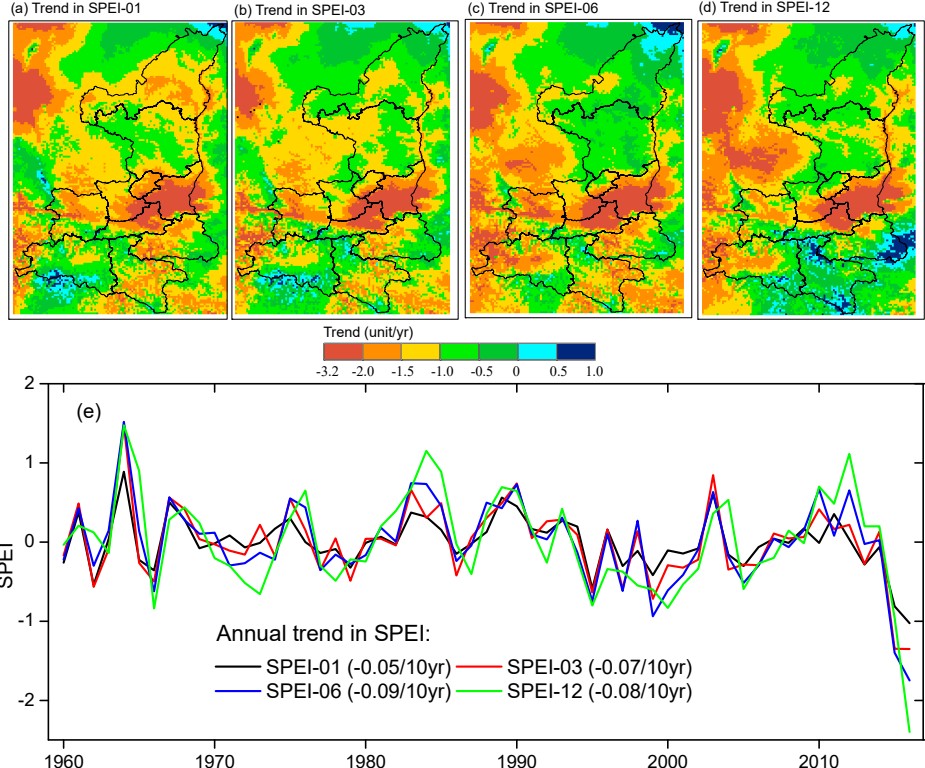

**Figure 5.** Spatial patterns in the annual trend of SPEI at 1-, 3-, 6-, and 12-month lags (**a–d**), and inter-annual variations and annual trends in SPEI time series at four typical timescales (**e**).

### 3.3. Response of Crop Yield to Different Timescales of Drought

Here, we quantified the response of wheat and maize production to droughts by a correlation analysis. We calculated the Spearman's correlation coefficients between the monthly detrended SPEI time series during 1- to 12-month lags and the SYRS of winter wheat and summer maize from 1989 to 2016 (Figure 6). As a more negative SPEI value corresponds to more severe drought, a higher correlation coefficient indicates that the effect of drought on crop yields is greater. It should be noted that the correlation between the SPEI and winter wheat is calculated from October of the last year to June of the following year because it is overwintering crop, whereas the correlation between the SPEI and summer maize is calculated from June to October of the current year.

As shown in Figure 6, there is a significant difference in the response of yield for winter wheat and summer maize to different timescales of droughts. The wheat yield anomalies are most correlated ($r = 0.56$) with the SPEI at six-month timescales occurring in March, followed by the SPEI-05 occurring in April, which often corresponds to the stem elongation to heading stages of plant growth [50]. It

implies that the medium-term droughts in March contribute the most to yield losses in that region. Similar conclusion has also been reached by Kang et al. (2002) in Shaanxi Province [50], they designed a series of soil water deficit experiments and applied them at different growth stages of winter wheat; they found that the severe water stress during stem elongation to heading stages significantly reduces the yield of winter wheat. Moreover, a less correlation between the SPEI and SYRS was detected in the maturity stage of winter wheat (May to June), where excessive precipitation may also have a negative influence on crop yield. The summer maize traditionally planted after harvesting winter wheat, and its yield anomalies are more sensitive to short-term droughts and show the most strongly correlated (*r* = 0.47) with the three-month SPEI (Figure 6b). This is different from the winter wheat: the yields of winter wheat are more sensitive to the medium-term droughts (SPEI-06, see Figure 6a). This difference may because the summer maize has a shorter growth period and the greater daily water demand than winter wheat. Further, the maximum correlation coefficient between the SPEI-03 and SYRS of summer maize was recorded in August, corresponding to the silking stage of summer maize. This finding is consistent with the conclusion reached by Liu et al. (2018) [33] in the North China Plain.

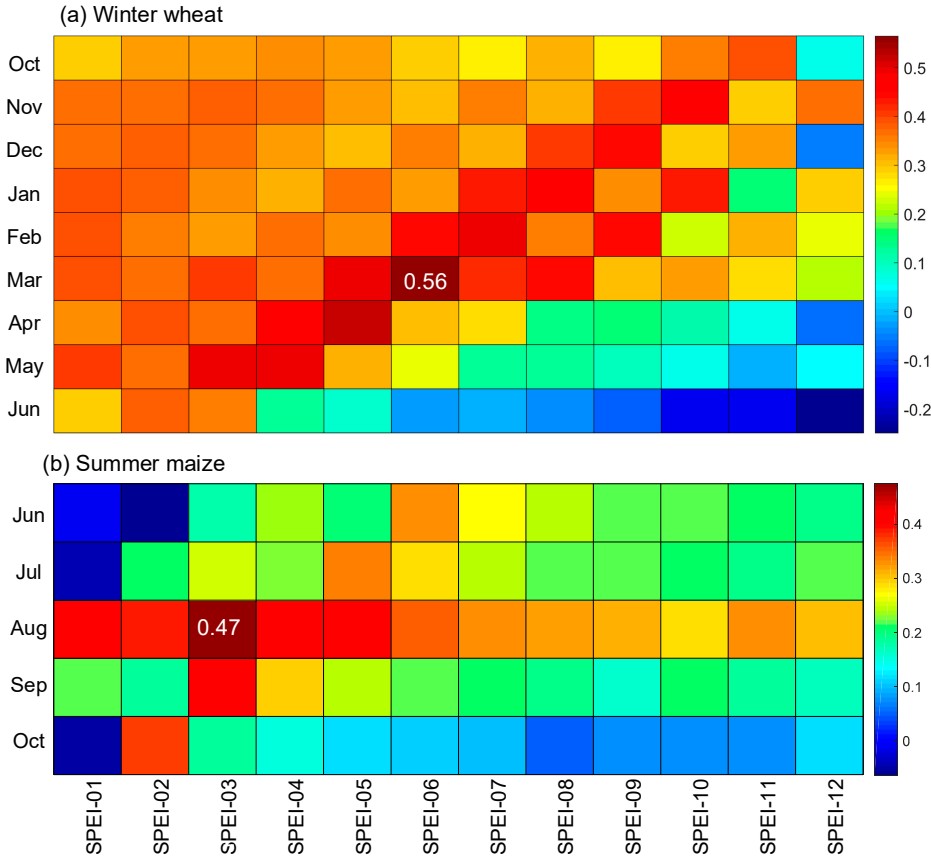

**Figure 6.** Spearman's correlation coefficients (*r*) between the monthly detrended SPEI during 1- to 12-month lags and the SYRS of winter wheat (**a**) and summer maize (**b**) from 1989 to 2016. The number marked with the white font denotes the maximum correlation between the detrended SPEI and SYRS of winter wheat or summer maize.

### 3.4. Assessment of Agricultural Drought Risk

As mentioned above (see the Section 2.3), the assessment of agricultural drought risk includes the assessment of the hazard and the vulnerability, which are then combined to form a comprehensive index of agricultural drought risk (DRI). We assessed the agricultural drought hazard index (DHI) based on the frequency and weight of each type of drought. As shown in Figure 7, the low risk dominates the spatial pattern of DHI at four typical timescales. The level of hazard of a region depends

on the timescale of SPEI used. For SPEI-01, the areas with high and very high hazard are mainly located in the northeastern (e.g., the junction area of Yulin and Yan'an) and western (e.g., Baoji and Xianyang) parts of Shaanxi Province. While the high and very high hazard based on SPEI-12 primarily appears in the north of the province. The SPEI-12 based DHI produces the largest area with high and very high risk, accounting for 20.5% of province's total area, followed by SPEI-01 (17.4%), SPEI-06 (9.6%), and SPEI-03 (5.8%).

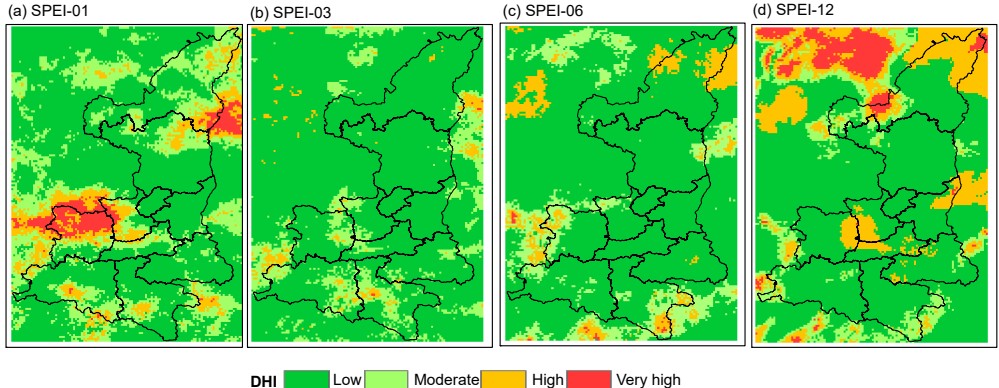

**Figure 7.** Spatial patterns of drought hazard index for four typical timescales of SPEI, i.e., the SPEI-01 (**a**), SPEI-03(**b**), SPEI-06 (**c**), and SPEI-12 (**d**).

The spatial distribution of the agricultural drought vulnerability index (DVI) and the used assessment indicators are shown in Figure 8. The three-vulnerability indicator were combined to produce a composite index of agricultural drought vulnerability (DVI), as shown in Figure 8c, which reflects the resilience of agriculture to cope with the consequences of droughts. Overall, the distribution of DVI is strongly affected by the climatic factor (i.e., aridity index), showing a decreasing trend from north to south of the province. However, the irrigation condition and soil properties also play an important role in the distribution of DVI in areas with a high percentage of the irrigated area or a small AWC. The areas with low vulnerability are concentrated in the central plains and southern mountains of the province; while the areas with a high and very high vulnerability are primarily located in the northern part of the province. The areas with a low, moderate, high, and very high vulnerability as a percentage of the province's total area are 21.2%, 32.5%, 30%, and 16.3%, respectively.

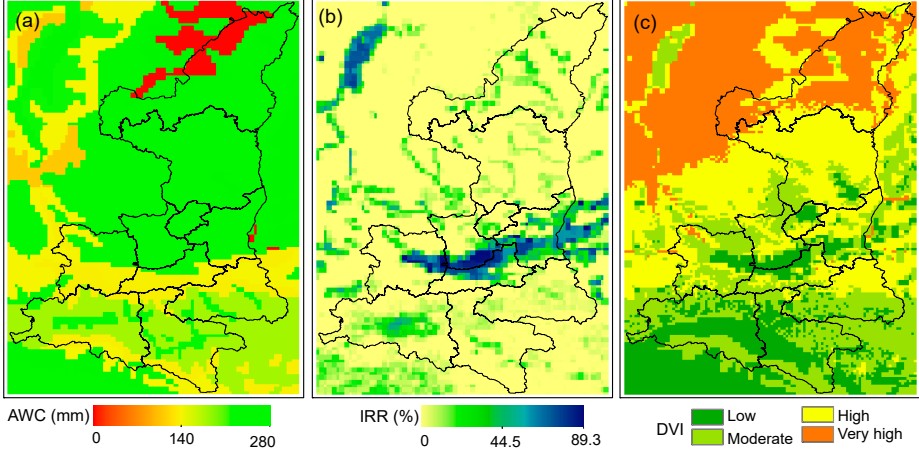

**Figure 8.** Grid-based maps of drought vulnerability indicators: (**a**) available soil water-holding capacity (AWC, mm) and (**b**) the percentage of area equipped for irrigation (IRR, %). (**c**) The spatial pattern of agricultural drought vulnerability index (DVI). The assessment indicator AI (aridity index) has been shown in Figure 1b and thus is not presented in this figure.

Figure 9 shows the spatial pattern in a composite index of agricultural drought risk at four typical timescales across the study area. Although the spatial distribution of drought risk is affected by the timescale of SPEI, the results consistently indicate that some areas in the north of the province are exposed to a high and a very high risk of drought, and other regions are dominated by low risk. Moreover, some areas in the west of the province are vulnerable to the short-term (SPEI-01) drought risk (Figure 9a). Maps of DRI generated based on four typical timescales of SPEI show a significant difference in the areas with a higher risk. The map of DRI generated based on SPEI-01 has the largest areas with a higher risk and these areas account for 31.4% of the total area of the whole province, followed by SPEI-12 (12.7%), SPEI-03 (7.9%), and SPEI-01 (5.1%).

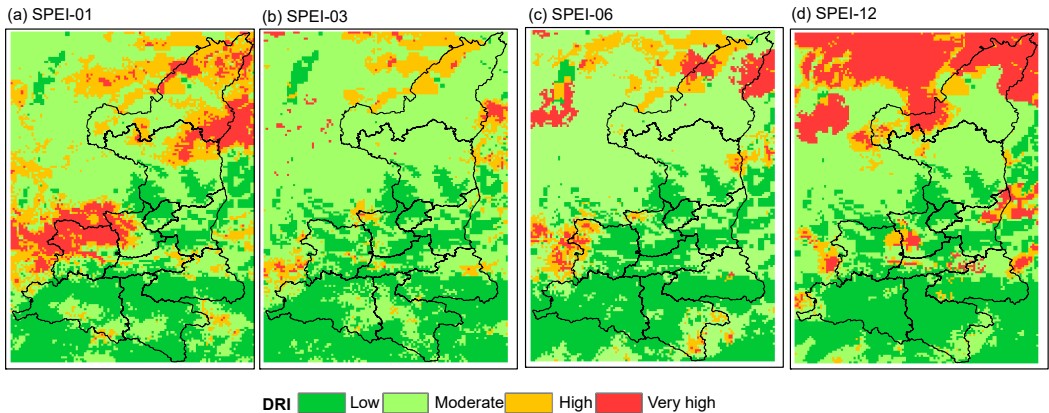

**Figure 9.** Spatial patterns in drought risk index (DRI) under four typical timescales of SPEI: the SPEI-01 (**a**), SPEI-03 (**b**), SPEI-06 (**c**), and SPEI-12 (**d**).

### 3.5. Features and Limitations

This study has three important features compared to previous studies. First, the PET is one of the important inputs for SPEI calculation and PET estimates from different methods differ greatly from each other [51]. Some studies have employed the temperature-based methods of PET estimates for SPEI calculation due to the nature of simplicity for these methods [21,23,52]. However, the temperature-based methods ignore the effects of radiation, humidity, and wind speed on PET estimates and respond only to changes in temperature. This simplification is likely to overestimate the drought trend and intensity under global warming [53]. In contrast, the Penman-Monteith method has solid physical meanings and considers both radiation and the aerodynamic component affecting PET. It has long been used as a benchmark to assess other PET methods. Second, we analyzed the correlation between the detrended SPEI and yield anomaly time series based on the actual growing season of crops instead of the whole calendar years of crops. This can more objectively reflect the sensitivity of crop yields to droughts. Third, numerous assessments of agricultural drought risk have been conducted both inside and outside of China [23,24,41]. However, to the best of our knowledge, this study was the first to assess agricultural drought risk in Shaanxi Province. The assessment results clearly show us where agricultural production in the province is susceptible to drought damage.

There are two potential limitations to apply the SPEI for agricultural drought assessment. First, the SPEI is an indicator of meteorological drought and therefore may not fully reflect the actual water stress experienced by crops. Second, the SPEI may overestimate the drought intensity during the early stage of crop growth season because the SPEI uses the PET instead of actual evapotranspiration to represent the atmospheric water deficit [54]. Furthermore, the assessment model of agricultural drought risk used here involves a larger number of weights and ratings to be assigned. The values of weights and ratings were determined based on the experiences of model developers. Whether these values can truly reflect the risk of drought is still debatable.

## 4. Conclusions

Drought research is currently a hot topic, especially in the context of global warming. Numerous studies on the assessment of drought evolution have been performed at a regional or global scale [45,47,53]. However, few studies focused on the drought impacts on agriculture. In this study, we assessed the spatiotemporal evolution of droughts between 1960 and 2016 in Shaanxi Province and investigated the influence of droughts on yield losses of winter wheat and summer maize. The spatial characteristics of agricultural drought risk were also assessed across the province. The main conclusions are summarized as follows.

(1)    The droughts have become more serious and frequent in Shaanxi Province after 1995, and the most severe drought episodes occurred in 2015–2016. On a spatial pattern, the most significant decreasing trend appears in the central plains of the province, where is the main producing area of the province's grain.

(2)    Medium-term droughts (quantified by SPEI-06) that occurred in March contribute the most to yield losses for winter wheat; yield loess of summer maize is more sensitive to the short-term drought (quantified by SPEI-03) that occurred in August.

(3)    Despite the spatial distribution and extent of agricultural drought risk are affected by the timescale of SPEI, some areas in the north of the province are exposed to the higher agricultural drought risks than other regions.

This research highlights that the response of crop yield to drought is affected by the timing of drought. The sensitivity of crop yield to drought is not the same in different growing stages of crops. Several studies have reported that moderate drought stress in specific growth stages of crops has little influence on crop yields [50,55]. To minimize the impact of drought on crop yields under limited water resources conditions, agricultural irrigation should give priority to the crop water demand during the period that the crop is most sensitive to drought.

**Author Contributions:** Formal analysis, Yongsheng Wang; Software, Qiubo Long; Writing–original draft, Mengjing Guo; Writing–review & editing, Jing Li and Peng Bai.

**Funding:** This research was supported by the Natural Science Foundation of China (No.41601034, 41807156, and 51979263). The authors appreciate the Editor and the anonymous reviewers for their efforts and constructive comments on the manuscript.

**Conflicts of Interest:** The authors declare no conflicts of interest.

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
