# Peer review of "Spatiotemporal Variations of Meteorological Droughts and the Assessments of Agricultural Drought Risk in a Typical Agricultural Province of China"

_atmosphere, doi:10.3390/atmos10090542_

Round 1

Reviewer 1 Report

This is a very interesting research and the method is sound. I think the research is worth of being published. In order to make it better, I think the authors should address following issues. 

First, I think it is interesting to focus on wheat and maize. But, why the study choose these two crops? I think the authors should discuss more clearly on their research design. 

Second, the authors used SPEI model. Could the authors introduce some similar studies using SPEI model? If it is new, please compare SPEI model with the existing models. 

Third, according to Figure 4, the drought is not evenly distributed. The authors should not only consider the temporal scale, but also the spatial scale. 

Fourth, how the inter-regional interactions could affect the crop production? Did authors consider the policies of China, or Shaanxi Province? 

Fifth, the authors should consider to proofread English by some native speaker. 

I hope my comments could be helpful to improve the manuscript. 

Author Response

Thanks very much for your insightful comments and these suggestions certainly have contributed to improve our manuscript. The response to each comment was listed as below.

Point 1: First, I think it is interesting to focus on wheat and maize. But, why the study choose these two crops? I think the authors should discuss more clearly on their research design. 

Response 1: suggestion is taken. We explained the reason for choosing wheat and maize for research. Please see below.

Lines 11-115 “The annual double-crop rotation system, winter wheat and summer maize, is the most popular planting pattern in the province. The planting area of winter wheat and summer maize accounts for more than 70% of the province's cultivated land area, and their output accounts for about 85% of the province's total grain output[26].”

Point 2: Second, the authors used SPEI model. Could the authors introduce some similar studies using SPEI model? If it is new, please compare SPEI model with the existing models. 

Response 2: suggestion is taken. The SPEI was not a new model, which has proposed by Vicente-Serrano et al. (2010) and was widely used for drought assessment. As your suggestion, we added some similar studies using SPEI for drought research (see below).

Lines 80-83: “For example, Labudová, et al. [19] reported that the SPEI reaches the higher correlation with the yield of maize than the SPI in Slovakia. Vicente-Serrano, et al. [22] found that the SPEI performs better than SPI and PDSI in identifying drought impacts on hydrological , agricultural, and ecological response variables at the global scale.”

Lines 156-161: “In general, the SPEI values at shorter timescales (e.g., 1-month and 3-month) reflect the short-term moisture conditions and often has a good correlation with soil moisture; while the SPEI values at longer timescales (e.g., 6-month and 12-month) reflect the medium- and long-term water balance patterns and are usually correlated to river flows, reservoir storages, and groundwater levels [23]. In practices, the SPEI-03 is commonly used to characterize agricultural droughts since it is more correlated with soil moisture than other timescales of SPEI [19,29,30].”

Point 3: Third, according to Figure 4, the drought is not evenly distributed. The authors should not only consider the temporal scale, but also the spatial scale. 

Response 3: suggestion is taken.

Point 4: Fourth, how the inter-regional interactions could affect the crop production? Did authors consider the policies of China, or Shaanxi Province? 

Response 4: as your point, the crop yield is affected by many factors, including agricultural policies, inter-regional interactions, advances in agricultural technology, grain price, fertilizer application, and droughts. Thus, the collected crop yield data cannot be directly used to establish a drought–yield relationship. To remove the effect of these non-climatic factors, we applied the detrending method to eliminate the trend in annual yield time series and maintained the inter-variations in yield time series. Then, the drought impact on the fluctuation of the trended annual yield time series was quantified by the correlation analysis. We provided more information on how to remove the influences of these non-climatic factors on crop yield, please see below for detains.

Lines 169-175: “The collected crop yield data cannot be directly used to establish a drought–yield relationship because the change in crop yield is affected by several factors in addition to droughts, such as new management practices, advances in agricultural technology, grain price, and increased fertilizer application. To remove the effect of these non-climatic factors, the SYRS was utilized to perform the correlation analysis with the detrended SPEI time series. The SYRS eliminates the trend in annual yield time series but maintains the inter-annual variation of the yield time series [19,21,31]”

Point 5: Fifth, the authors should consider to proofread English by some native speaker.

Response 5: suggestion is taken. The manuscript has been polished by a professional English editing agency (www.aje.com).

Reviewer 2 Report

In figure 4, the color scheme suggests that red colors indicate lower frequency of drought and blue colors indicate higher frequency of drought? If so, consider flipping the color bar so that red colors are higher frequency of drought -- this is more intuitive. If that is not the case, the color bar title and caption can be improved to better explain the figures. Is the frequency given in percent of years? 

Likewise in figure 9, the authors should consider a color scheme that is more intuitive. Typically reds, oranges, and yellows indicate drought/dryness and blues and greens indicate wetness/precipitation. I recommend at a minimum inverse color scheme so that high drought risk is associated with red/yellow colors and low drought risk is associated with blue colors. Perhaps the authors could consider the spectral, red-yellow-blue, or red-yellow-green diverging color maps here https://www.mathworks.com/matlabcentral/mlc-downloads/downloads/submissions/34087/versions/2/screenshot.jpg

The primary comment I have is that the authors should consider adding to their Conclusions section perhaps 2 or 3 paragraphs that place their results within the context of other published literature, and provide some discussion on the importance of their findings. That is, why should the scientific community care about the results of this work. This may be in part re-framing parts of the introduction, but in the context of the results that have now been presented. However, if this type of conclusions section is not common in this particular field of study or in this journal, then the authors may choose to disregard this comment. 

Author Response

Thanks very much for your insightful comments and these suggestions certainly have contributed to improve our manuscript. The response to each comment was listed as below.

Point 1: In figure 4, the color scheme suggests that red colors indicate lower frequency of drought and blue colors indicate higher frequency of drought? If so, consider flipping the color bar so that red colors are higher frequency of drought -- this is more intuitive. If that is not the case, the color bar title and caption can be improved to better explain the figures. Is the frequency given in percent of years? 

 Response 1: suggestion is taken. We have revised the color schemes by flipping the color bar. Please see the revised manuscript for details.

Point2: Likewise in figure 9, the authors should consider a color scheme that is more intuitive. Typically reds, oranges, and yellows indicate drought/dryness and blues and greens indicate wetness/precipitation. I recommend at a minimum inverse color scheme so that high drought risk is associated with red/yellow colors and low drought risk is associated with blue colors. Perhaps the authors could consider the spectral, red-yellow-blue, or red-yellow-green diverging color maps here https://www.mathworks.com/matlabcentral/mlc-downloads/downloads/submissions/34087/versions/2/screenshot.jpg

 Response 2: suggestion is taken. We adjusted the color scheme of figure 9 as your suggestion. In the revised image, the four types of risk, that is, the low, moderate, high, and extreme risk, were indicated by blue, green, orange, and red, respectively.

Point3: The primary comment I have is that the authors should consider adding to their Conclusions section perhaps 2 or 3 paragraphs that place their results within the context of other published literature, and provide some discussion on the importance of their findings. That is, why should the scientific community care about the results of this work. This may be in part re-framing parts of the introduction, but in the context of the results that have now been presented. However, if this type of conclusions section is not common in this particular field of study or in this journal, then the authors may choose to disregard this comment. 

Response 3: suggestion is taken. As your suggestion, we added the following contents in the Conclusions section, please see below for details.

Lines 398-400: “Drought research is currently a hot topic, especially in the context of global warming. Numerous studies on the assessment of drought evolution have been performed at a regional or global scale [41,43,49]. However, few studies focused on the drought impacts on agriculture.”

Lines 415-420: “This research highlights that the response of crop yield to drought is affected by the timing of drought. The sensitivity of crop yield to drought is not the same in different growing stages of crops. Several studies have reported that moderate drought stress in specific growth stages of crops has little influence on crop yields [46,51]. To minimize the impact of drought on crop yields under limited water resources conditions, agricultural irrigation should give priority to the crop water demand during the period that the crop is most sensitive to drought than other periods.

Reviewer 3 Report

Dear Authors,

the reviewed article entitled "Spatiotemporal variations of meteorological droughts and the assessments of agricultural drought risk in a typical agricultural province of China" corresponds well with the current trend of research in climatology. The article requires several corrections before publication.

Fig. 1. the station numbers are not visible. The indicators used should be described in more detail. In the description of Fig. 2. there is the designation "a" and "b", and this is not in Fig. Figs. 5a-d are not equal. Similarly, Figs. 7 and 9.

Author Response

Point 1: Fig. 1. the station numbers are not visible. The indicators used should be described in more detail.

Response 1: The Figure 1 was distortion when the manuscript in the word version was transferred to PDF version. In the revised manuscript, we have solved this problem.

Point2: In the description of Fig. 2. there is the designation "a" and "b", and this is not in Fig.

Response 2: we have removed the designation "a" and "b" .

Point 3: Figs. 5a-d are not equal. Similarly, Figs. 7 and 9.

Response 3: we rewritten the captions of these figures. Please see below. 

Figure 5. Spatial patterns in the annual trend of SPEI at 1-, 3-, 6-, and 12-month lags (a-d), and inter-annual variations and annual trends in SPEI time series at four typical timescales (e).

Figure 7. Spatial patterns of drought hazard index for four typical timescales of SPEI, that is, the SPEI-01 (a), SPEI-03(b), SPEI-06 (c), and SPEI-12 (d).

Figure 9. Spatial patterns in drought risk index (DRI) under four typical timescales of SPEI: the SPEI-01 (a), SPEI-03(b), SPEI-06 (c), and SPEI-12 (d).

Round 2

Reviewer 1 Report

I have no more comments.